# Effect of Mat Pilates on Body Fluid Composition, Pelvic Stabilization, and Muscle Damage during Pregnancy

## Ah-Hyun Hyun [1] and Yoo-Jeong Jeon [2,*]

[1] Department of Health and Exercise Science, Korea National Sport University, 1239, Yangjae-daero, Songpa-gu, Seoul 05541, Korea; knupe838@knsu.ac.kr

[2] Department of Sports & Health Science, Hanbat National University, Yoo Seong-gu 125, Daejeon 34158, Korea

[*] Correspondence: jyjong20@hanbat.ac.kr

**Abstract:** In this study, according to the exercise intensity (50–60% of HRmax (Maximum Heart Rate), RPE (Rating of Perceived Exertion: 11–13) proposed by The American Congress of Obstetricians and Gynecologists (ACOG) for pregnant women, mat Pilates exercise is related to body composition, lipid parameters, and pelvic stabilization. The effects on muscle and muscle damage were investigated. The subjects of this study were 16 pregnant women registered at the Cultural Center of Gyeonggi-do C Women's Hospital, and the gestation period was 16 to 24 weeks. The mat Pilates exercise program (twice a week, 60 min per day, total 12 weeks) changed the Pilates exercise program every 6 weeks according to the subject's pain level and physical fitness. Body composition before and after exercise, hip flexion, abduction and dilated lipids, inflammation, muscle damage, and stress hormones were measured through blood biochemical analysis. First, the difference in total body water, intracellular water, and skeletal muscle changes (post-pre) increased significantly in the Pilates exercise (PE) group compared to the control (CON) group, while the extracellular/intracellular water ratio significantly decreased. The effect of Pilates on body composition and lipid profile confirmed that, after testing, total body water (TBW), intracellular water (ICW), and extracellular water (ECW) were significantly greater than pre-test values in both groups (TBW: $z = -2.286$, $p = 0.022$, $r = 0.572$; ICW: $z = -2.818$, $p = 0.005$, $r = 0.705$; ECW: $z = -1.232$, $p = 0.218$, $r = 0.308$), whereas the ECW/ICW ratio decreased significantly only in the PE group ($z = -2.170$, $p = 0.030$, $r = 0.543$); while the increases in TBW and ICW were greater in the PE group than in the CON group, the ECW/ICW ratio decreased significantly in the PE group. Blood tests showed significant increases in body weight (BW), body fat mass (BFM), and percentage of body fat (PBF) in both groups post-test as compared to pre-test (BW: $z = -1.590$, $p = 0.112$, $r = 0.398$; BFM: $z = -0.106$, $p = 0.916$; PBF: $z = -1.643$, $p = 0.100$, $r = 0.411$). There was a slight increase in creatine kinase (CK) and lactate dehydrogenase (LDH), which are indices of muscle damage, and in the difference between the periods within the group, the CK and LDH of the CON group showed a tendency to increase significantly after inspection compared to the previous values (CK: $z = -1.700$, $p = 0.089$, $r = 0.425$, LDH: $z = -2.603$, $p = 0.009$, $r = 0.651$). Aspartate aminotransferase (AST) decreased significantly in the Pilates exercise group compared to that in the control group, and as a result of confirming the difference in the amount of change in C-reactive protein (CRP), there was no significant difference between the two groups, and the PE group showed a tendency to decrease after inspection compared to the previous period even in the difference between the periods in the group. The CON group showed an increasing trend, but no significant difference was found. Cortisol, a stress hormone, also increased significantly after inspection both groups compared to before (CON group: $z = -2.201$, $p = 0.028$; PE group: $z = -2.547$, $p = 0.011$). Therefore, the 12 week Pilates exercise program conducted in this study has a positive effect on body water balance and strengthens the muscles related to pelvic stabilization within the range of reducing muscle damage or causing muscle damage and stress in pregnant women. We think that it has an effective exercise intensity.

**Keywords:** exercise intensity; body composition; lactate dehydrogenase; stress hormone

---

## 1. Introduction

In general, pregnancy causes a variety of anatomical, physiological, and mental changes in women [1]. They experience physical changes such as low back pain, pelvic pain, and swelling due to weight gain and body shape change and feel joint pain due to spasms and hormone secretion [2]. In the latter half of pregnancy, there is the additional weight of the fetus and amniotic fluid, and women experience a great deal of physical stress and suffer from insomnia [3,4]. Consequently, failure to adapt to physical changes during pregnancy may lead to gestational depression, and the mother's mental stress has a negative effect on the fetus [5–7]. Therefore, efforts to manage the series of changes caused by pregnancy are essential. Methods such as massage, nutrition, music, and psychological counseling have been suggested to reduce the physical and mental stress of pregnant women, and the American Congress of Obstetricians and Gynecologists (ACOG) recommends physical activities, including stretching [8–10].

Participation in exercise for pregnant women is effective in managing weight, improving constipation and cardiopulmonary function, reducing neck pain, low back pain, and pelvic pain, and preventing cesarean delivery [11–13]. Exercises recommended by the ACOG include walking, swimming, and indoor cycling, and it has been established that most pregnant women choose walking as a physical activity that they can perform without burden [14–17]. However, walking has a risk of falling due to a change in the center of gravity and weight gain during pregnancy and is considered to be functionally insufficient to improve physical strength for childbirth and to reinforce weak muscle strength due to pregnancy. In addition, since the application of exercise programs that do not take into account individual characteristics and pregnancy status may negatively affect the mother or fetus, safe guidelines for exercise programs during pregnancy are essential. In this regard, it has recently been suggested that Pilates exercise is safe for pregnant women and can stimulate the muscles around the core and pelvis, and the ACOG also actively recommends Pilates exercise for pregnant women [18]. Pilates exercise has been suggested to increase muscle strength, flexibility, coordination, and pelvic stability in the form of a combination of aerobic and anaerobic exercise [19–21], it especially improves physical discomfort in pregnant women and is effective for weight control and back pain [22–25].

Previous studies on pregnant women suggested that regular Pilates improves breathing ability, muscle strength, and postural stability and is effective in improving basic physical strength such as cardiopulmonary function through whole-body exercise [25,26]. In addition, resistance exercise using props increases muscle strength and balance ability to prevent falls, increases basal metabolism, and reduces pain and fatigue at the end of pregnancy [25]. However, contradictory studies have reported that Pilates is not effective in improving the core muscles of pregnant women [27,28]. This is probably due to the lack of previous studies demonstrating the effectiveness of Pilates exercise for pregnant women and the lack of various data on the intensity and frequency of appropriate types of exercise for pregnant women [29].

Particularly, since repetitive Pilates movements and excessive muscle use in pregnant women have the potential to cause more muscle damage as well as damage to joints [30], the Pilates exercise program for pregnant women should be carefully selected. Creatine kinase (CK) is an index indicating the degree of muscle damage. The blood CK concentration of pregnant women before 34 weeks is correlated with gestational hypertension [31], and the level is high depending on the amount of activity with high exercise intensity [32,33]. Previous studies have suggested that Pilates exercise using resistance tools directly stimulates the muscles to induce muscle damage and increases CK concentration [30,31], whereas the exercise group in the 12 week yoga exercise group compared unfavorably to the control group. It was suggested that the level of inflammation markers decreased, thereby reducing inflammation [34]. As such, the different results of the CK study are due to the fact

that the results of muscle injury vary greatly depending on the intensity of the exercise. In particular, studies on Pilates exercise and CK studies in pregnant women are insufficient, and additional verification is needed. Therefore, it is necessary to first evaluate the exercise experience and physical ability of the pregnant woman before pregnancy, collect the mother's information, and then divide and program the exercise intensity according to the pregnancy week. Therefore, in this study, according to the exercise guidelines proposed by the ACOG (50~60% of the maximum heart rate) and exercise awareness using the Borg Scale (11~13), 12 weeks of Pilates exercise is used [35]. The purpose of this study is to analyze the effects of body composition, lipid variables, and muscle and muscle damage related to pelvic stabilization to confirm the intensity and effect of Pilates exercise during pregnancy.

## 2. Materials and Methods

### 2.1. Participants

The subjects of this study were 16 pregnant women registered at the Cultural Center of C Women's Hospital in Gyeonggi-do, South Korea, who did not have specific diseases or receive specific medications, had no pregnancy complications, and were expected to have a normal delivery (Gestation period: 16~24 weeks, PE group: n = 9, CON group: n = 7). Prior to conducting the study, all subjects were fully aware of the prior explanations, and voluntary participation and consent were ascertained before conducting the study.

This study was a cluster-randomized controlled experiment. A total of 16 subjects were recruited to the Pilates exercise group (PE, n = 9; age: 31.78 ± 4.68 years; body weight (BW): 57.97 ± 7.91 kg; height: 160.56 ± 3.78 cm, using the block randomization method; body fat mass (BFM): 18.40 ± 4.92 g; body mass index(BMI): 39.57 ± 4.52; skeletal muscle mass (SMM): 21.10 ± 2.66 g)), and the control group participants (CON, n = 7; age: 32.00 ± 3.46 years; BW: 53.67 ± 5.70 kg; height: 161.29 ± 3.54 cm; BFM: 15.86 ± 3.06 g; BMI: 37.81 ± 3.37; SMM: 20.09 ± 2.00 g) were randomly assigned.

A comparison of the reference values between groups was performed by obtaining Cohen's r values. The research protocol was approved by the Bioethics Committee of Korea National Sports University (1263-201803-BR-001-01).

### 2.2. Pilates Exercise Protocols

The Pilates exercise program consisted of warm-up, main, and cool-down exercises and was conducted for 60 min per day, twice a week, for 12 weeks. The exercise intensity was set as 50–60% of maximum heart rate, as suggested by the ACOG [23], and the Borg Scale was implemented during the exercise to maintain an RPE(Rating of Perceived Exertion) of 11~13. Depending on the participants' level of pain and physical fitness, the intensity of the Pilates exercise was progressively increased every 6 weeks (Table 1).

### 2.3. Pelvic Stabilization Muscle Strength Test

To measure the strength of muscles involved in pelvic stabilization, hip flexion (HF), hip abduction (HA), and hip extension (HE) were measured. Considering that the participants were pregnant women, muscle strength was measured after sufficient explanation and 1–2 practice sessions. Approximately 1 min was provided for rest for each area of measurement, and a trained physical therapist made the measurements in a stable position that did not cause any pain. HF was measured using the de Groot active straight leg raising (ASLR) test method, with both hands lowered to the floor and both feet shoulder-width wide in a straight position [36]. The examiner placed a manual muscle strength tester (HOGGAN PROOF Preferred, HOGGAN HEALTH, USA) on the participant's right ankle, and the participant was asked to raise the leg being measured toward the ceiling completely to measure the muscle strength.

**Table 1.** Pilates exercise program.

| Modes | Week | Contents | Time (min) | Reps, Set, and Rest | RPE |
|---|---|---|---|---|---|
| Warm-up | 1–12 | (Breathing · Static Stretching) Neck and Shoulder Stretch, Deep Breathing Leg Stretch | 10 | | 9 |
| Main Exercise | 1–6 (Level 1) | Torso Twist, Cat Cow, Kneeling Half Push-up Lying One-Leg Circles One-Leg Side Kick, Pelvic Stretch | 30 | 8~12 reps × 3 set 20 s rest between sets | 9~11 |
| | 7–12 (Level 2) | Spine Stretch, Double-Arm Circles Half Roll Down and Up, One-Leg Side Rotation Lying Leg Scissors, Donkey Kick | 30 | 8~12 reps × 3 set 30 s rest between sets | 12~13 |
| Cool-down | 1–12 | (Breathing · Static Stretching) Neck and Shoulder Stretch Deep Breathing | 10 | | 9 |

To measure HA, the participant was asked to lie down in a lateral decubitus position with the head resting comfortably on one arm and one foot on top of the other, and the examiner placed the manual muscle strength tester on the ankle on the top. Subsequently, the participant was asked to raise the side of the hip joint completely toward the ceiling to measure the HA strength. Lastly, to measure HE, the participant was asked to stand with her legs shoulder-width apart while placing her hands on the wall. The examiner placed the manual muscle strength tester on the participant's right ankle, and the participant was asked to completely raise the right leg posteriorly to measure the HE strength. Three measurements were made for all variables, and the mean of the measurements was used for analysis. The unit of measurement was 0.45 kg, and the margin of error was ±1%.

### 2.4. Body Composition

BW, total body water (TBW), intracellular water (ICW), extracellular water (ECW), body fat mass (BFM), percent body fat (%BF), and skeletal muscle mass (SMM) were measured using eight-polar bioelectrical impedance analysis (BIA) with multiple impedance frequencies (Inbody 770, Biospace Co., Seoul, Korea). All participants were allowed to limit food intake and empty their bladder 4 h before measurement according to the instructions of the manufacturer. After that, it was measured using a fatness measuring system (Dong-Sahn Jenix, Korea). Each participant stood with her soles in contact with the foot electrodes and grabbed the hand electrodes, wearing light clothing and removing all metal items to ensure accurate body composition measurement.

### 2.5. Blood Collection and Biochemical Analyses

Blood samples were drawn from the antecubital vein via multiple venipunctures for the determination of Ferritin; HbA1c; cortisol; lipid-related markers, such as total triglycerides (TG), total cholesterol (TC), high-density lipoprotein (HDL), low-density lipoprotein (LDL); and muscle damage markers such as creatine kinase (CK), lactate dehydrogenase (LDH), C-reactive protein (CRP), and aspartate aminotransferase (AST). Blood samples were taken 12 h before and 12 h after the 12week intervention period and frozen at −80 °C to prevent denaturation and ensure stability of all analytes stored in a deep freezer (Nihon Freezer Co., Japan, VT-208) until analysis. All the assays were carried out according to the instructions of the manufacturers.

## 2.6. Statistical Analysis

SPSS 24.0 was used to analyze the pre- and post-test differences in body composition, strength of muscles involved in pelvic stabilization, and muscle damage markers after Pilates exercise. Due to the small sample size for the measured variables, normal distribution could not be assumed for the variables; therefore, all statistical analyses were conducted through non-parametric tests. For between-group differences, the differences in mean (the post-test mean minus the pre-test mean) obtained through change-score analysis were analyzed through Mann–Whitney U test, and within-group differences were analyzed through Wilcoxon signed-rank test. All statistics are presented as mean and standard deviation, and the level of significance for statistical analysis was set as $\alpha < 0.05$.

## 3. Results

### 3.1. Effect of Pilates on the Body Compositions and Lipid Profiles

In this study, confirmed post-test TBW, ICW, and ECW were significantly greater than pre-test values in both groups (TBW: $z = -2.286$, $p = 0.022$, $r = 0.572$; ICW: $z = -2.818$, $p = 0.005$, $r = 0.705$; ECW: $z = -1.232$, $p = 0.218$, $r = 0.308$; Table 2), whereas the ECW/ICW ratio decreased significantly only in the Pilates exercise (PE) group ($z = -2.170$, $p = 0.030$, $r = 0.543$; Table 2). While the increases in TBW and ICW were greater in the PE group than in the control (CON) group, the ECW/ICW ratio decreased significantly in the PE group.

**Table 2.** Body composition.

| | Pilates (*n* = 9) | | CON (*n* = 7) | | Diff (Post-Pre) | | |
|---|---|---|---|---|---|---|---|
| | **Pre** | **Post** | **Pre** | **Post** | **z** | **p** | **Cohen's r** |
| BW (kg) | 57.98 ± 7.91 | 64.01 ± 7.76 ** | 53.67 ± 5.70 | 58.37 ± 5.65 * | −1.590 | 0.112 | 0.398 |
| TBW (L) [#] | 28.97 ± 3.33 | 31.59 ± 3.50 ** | 27.63 ± 2.45 | 29.26 ± 2.84 * | −2.286 | 0.022 | 0.572 |
| ICW (L) [##] | 17.71 ± 2.06 | 19.54 ± 2.12 ** | 16.90 ± 1.54 | 17.99 ± 1.66 * | −2.818 | 0.005 | 0.705 |
| ECW (L) | 11.26 ± 1.28 | 12.04 ± 1.39 ** | 10.73 ± 0.92 | 11.27 ± 1.18 * | −1.232 | 0.218 | 0.308 |
| ECW/ICW ratio [#] | 0.64 ± 0.01 | 0.62 ± 0.01 * | 0.64 ± 0.01 | 0.63 ± 0.01 | −2.170 | 0.030 | 0.543 |
| BFM (kg) | 18.40 ± 4.92 | 20.70 ± 5.04 ** | 15.86 ± 3.06 | 18.30 ± 3.06 * | −0.106 | 0.916 | 0.027 |
| PBF (%) | 22.50 ± 3.15 | 24.84 ± 3.02 ** | 20.59 ± 1.64 | 22.41 ± 1.67 * | −1.643 | 0.100 | 0.411 |
| SMM (kg) [##] | 21.10 ± 2.66 | 23.49 ± 2.76 ** | 20.09 ± 2.00 | 21.41 ± 2.18 * | −2.811 | 0.005 | 0.703 |

BW: body weight, TBW: total body water, ICW: intracellular water, ECW: extracellular water, ECW/ICW: ECW to ICW ratio, BFM: body fat mass, PBF: percentage of body fat, SMM: skeletal muscle mass. Diff (post − pre): difference change from pre to post between groups. * $p < 0.05$, ** $p < 0.01$ from pre and post. [#] $p < 0.05$, [##] $p < 0.01$ change (post − pre) between groups. Values are presented as mean ± SD.

Blood tests showed significant increases in BW, BFM, and PBF in both groups post-test as compared to pre-test (BW: $z = -1.590$, $p = 0.112$, $r = 0.398$; BFM: $z = -0.106$, $p = 0.916$; PBF: $z = -1.643$, $p = 0.100$, $r = 0.411$; Table 2). Significant post-test increases in TG, TC, and LDL were also observed when compared to pre-test values (TG: $z = -0.106$, $p = 0.916$, $r = 0.027$; TC: $z = -1.059$, $p = 0.289$, $r = 0.265$; LDL: $z = -1.272$, $p = 0.203$, $r = 0.318$; HbA1a: $z = -0.530$, $p = 0.596$, $r = 0.133$; Table 3). In contrast, ferritin decreased significantly in both groups (Ferritin: $z = -0.530$, $p = 0.596$, $r = 0.133$) (Table 3).

### 3.2. Effect of Pilates on the Pelvic Stabilization Muscle Strength

In this study, we confirmed pre-test and post-test HF, HA, and HS. When changes in HF, HA, and HS were analyzed, they were found to have increased significantly in the PE group when compared to the CON group (HF: $z = -3.037$, $p = 0.002$, HA: $z = -3.344$, $p = 0.001$, HS: $z = -2.595$, $p = 0.009$; Figure 1A–C). Moreover, in terms of within-group differences, all measurements increased significantly post-test compared to the pre-test measurements in the PE group (HF: $z = -2.371$, $p = 0.018$; HA: $z = -2.670$, $p = 0.008$; HS: $z = -2.192$, $p = 0.028$), whereas only HA decreased significantly post-test in the CON group ($z = -2.375$, $p = 0.018$).

**Table 3.** Lipid profiles.

| | Pilates (*n* = 9) | | CON (*n* = 7) | | Diff (Post-Pre) | | |
|---|---|---|---|---|---|---|---|
| | **Pre** | **Post** | **Pre** | **Post** | **z** | **p** | **Cohen's r** |
| TG (kg) | 140.67 ± 35.25 | 201.56 ± 54.24 ** | 141.00 ± 38.59 | 205.86 ± 48.46 * | −0.106 | 0.916 | 0.027 |
| TC (mg/dL) | 220.33 ± 27.00 | 236.56 ± 25.93 * | 222.14 ± 28.54 | 248.29 ± 38.10 * | −1.059 | 0.289 | 0.265 |
| HDL (mg/dL) | 92.56 ± 11.67 | 90.11 ± 12.67 | 89.57 ± 13.62 | 85.00 ± 18.91 | −0.638 | 0.524 | 0.160 |
| LDL (mg/dL) | 110.67 ± 26.17 | 135.78 ± 25.25 * | 115.00 ± 18.06 | 152.00 ± 31.25 * | −1.272 | 0.203 | 0.318 |
| HbA1c (%) | 4.91 ± 0.18 | 5.13 ± 0.21 * | 4.91 ± 0.15 | 5.14 ± 0.15 * | −0.108 | 0.914 | 0.027 |
| Ferritin (ng/mL) | 55.44 ± 49.27 | 23.11 ± 11.41 * | 34.14 ± 6.72 | 16.57 ± 8.89 * | −0.530 | 0.596 | 0.133 |

Values are means ± SD. Main time effect: * $p < 0.05$ and ** $p < 0.01$ from pre and post. TC: total cholesterol, TG: triglycerides, LDL: low-density lipoprotein, HDL: high-density lipoprotein, HbA1c: hemoglobin A1c. Diff (post − pre): difference change from pre to post between groups. Values are presented as mean ± SD.

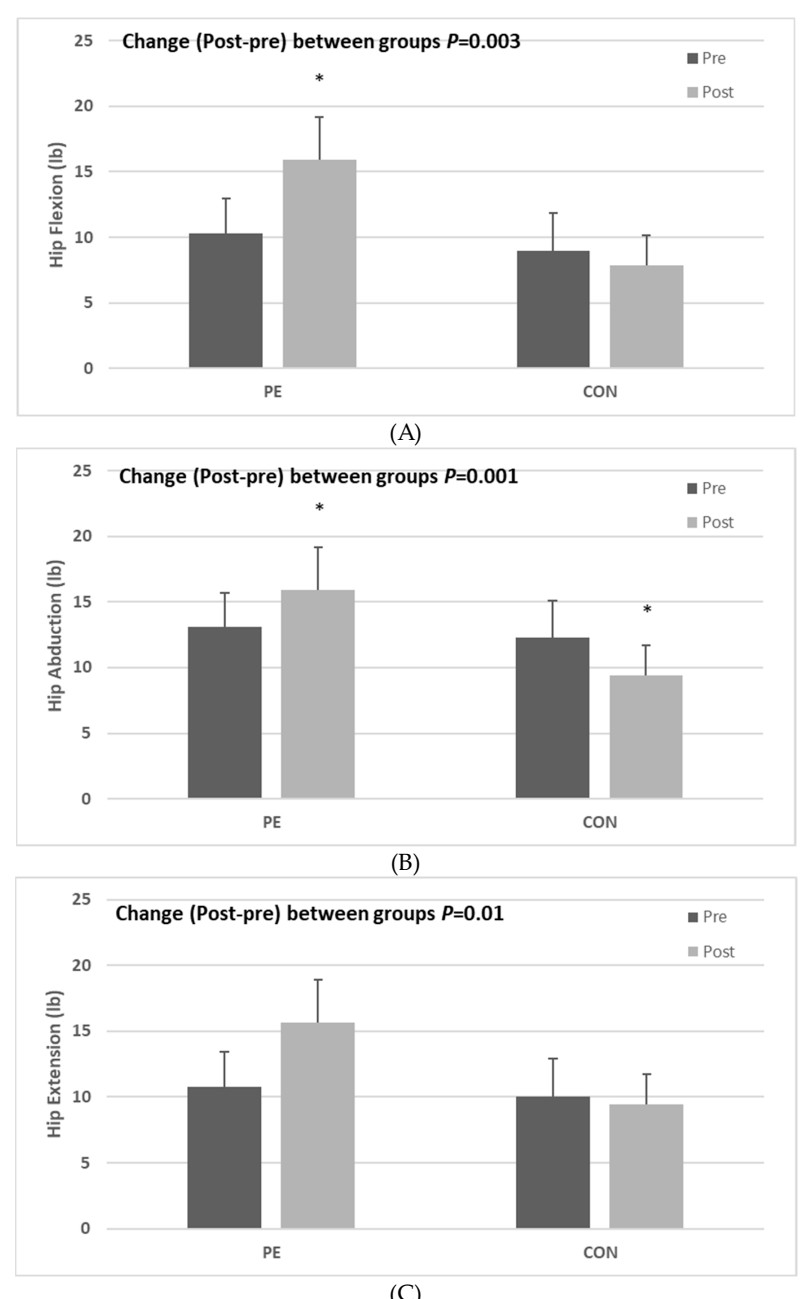

(A)

(B)

(C)

**Figure 1.** Effect of Pilates exercise on back pain-related muscle strength, (**A**) Hip flexion, (**B**) Hip abduction, and (**C**) Hip extension following 12 weeks of Pilates exercise. Bars represent mean ± SD (PE: n = 9, CON: 7, per group). * $p < 0.05$ from Pre to Post. PE: Pilates exercise group, CON: control group.

### 3.3. Muscle Damage and Stress Markers

In this study, the difference between CK and lactate dehydrogenase (LDH) changes was found to increase slightly in the PE group compared to that in the CON group (Table 4). In addition, differences in timing within the group indicated a significant post-test increase in CK and LDH for the CON group compared to that in the PE group (CK: $z = -1.700$, $p = 0.089$, $r = 0.425$; LDH: $z = -2.603$, $p = 0.009$, $r = 0.651$). AST changes were shown to have significantly decreased in the PE group compared to in the CON group (AST: $z = -2.566$, $p = 0.010$, $r = 0.642$) and showed a significant post-test increase in terms of timing in the CON group only. The difference in the amount of change in CRP, an indicator of inflammation, was not significant between the two groups (CRP: $z = -1.230$, $p = 0.219$, $r = 0.308$). Even in differences in timing within a group, PE groups tended to decrease after exercise compared to dictionaries, while CON groups tended to increase but with no significant differences (Table 4). The stress hormone cortisol also showed a significant post-test increase in both groups compared to the standard levels (cortisol: $z = -2.547$, $p = 0.011$, $r = 0.636$) (Figure 2), but there was no significant difference in the variation between the two groups.

**Table 4.** Muscle damage and inflammation marker.

| | Pilates (*n* = 9) | | CON (*n* = 7) | | Diff (Post − Pre) | | |
|---|---|---|---|---|---|---|---|
| | Pre | Post | Pre | Post | *z* | *p* | Cohen's *r* |
| CK (U/L) | 34.78 ± 8.66 | 42.22 ± 13.15 | 39.57 ± 16.06 | 59.43 ± 27.72 ** | −1.700 | 0.089 | 0.425 |
| LDH (U/L) ## | 159.00 ± 15.67 | 159.78 ± 15.09 | 162.29 ± 21.40 | 186.57 ± 26.79 ** | −2.603 | 0.009 | 0.651 |
| AST (U/L) # | 20.22 ± 6.32 | 17.78 ± 5.02 | 14.71 ± 4.11 | 16.29 ± 4.03 * | −2.566 | 0.010 | 0.642 |
| CRP (mg/L) | 0.25 ± 0.20 | 0.24 ± 0.23 | 0.10 ± 0.07 | 0.10 ± 0.06 | −1.230 | 0.219 | 0.308 |

Values are means ± SD. Main time effect: * $p < 0.05$ and ** $p < 0.01$ pre- versus post-Pilates period in the between groups. CK: creatine kinase, LDH: lactate dehydrogenase, AST: aspartate aminotransferase, CRP: C-reactive protein. * $p < 0.05$ from pre and post. # $p < 0.05$, ## $p < 0.01$ change (post − pre) between groups. Diff (post − pre): difference change from pre to post between groups. Values are presented as mean ± SD.

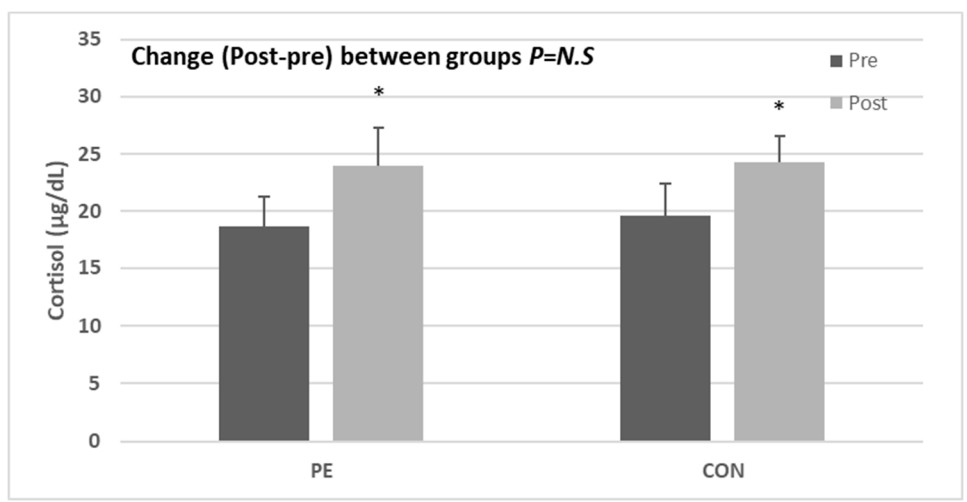

**Figure 2.** Effect of Pilates exercise on cortisol hormone level, bars represent mean ± SD (PE: n = 9, CON: 7, per group). * $p < 0.05$ from Pre to Post. PE: Pilates exercise group, CON: control group.

## 4. Discussion

### 4.1. Effect of Pilates on Body Compositions and Lipid Profiles

In general, as the number of weeks of pregnancy increases, most mothers have an imbalance in body fat and blood lipid metabolism, which can lead to hyperlipidemia and ischemic heart disease in the mother, so continuous management is required during pregnancy. In addition, sudden weight gain after pregnancy causes gestational hypertension and diabetes and negatively affects the microbial

environment in the intestine, leading to postpartum obesity [25,37]. Gestational edema, another characteristic that pregnant women experience, not only causes discomfort and pain but also decreases the growth of the fetus in the uterus and is deeply related to gestational hypertension [38].

In general, body hydration increases as the number of gestational weeks increases, and in particular, an increase in ECW causes edema [39]. The ACOG [40] reported an increase in ECW during pregnancy and that ECW and the ECW/ICW ratio are significantly higher in pregnant women with pre-eclampsia. Similarly, in this study, both groups significantly increased post TBW, ICW, and ECW compared to the standard values, while the ECW/ICW ratio decreased significantly only in the PE group.

Interestingly, the difference between TBW and ICW changes in the PE group was found to have increased over that in the CON group, while the ECW/ICW ratio was significantly reduced. These results may be thought to be partially mitigated by the increased rate of ECW that occurs during pregnancy and have shown some consistent results, with prior studies reporting a decrease in edema through Pilates exercise in cancer patients [41]. Moreover, the finding that Pilates exercise led to partial decreases in the ECW/ICW ratio, which increases in pre-eclampsia, suggests that Pilates exercise could be effective in relieving not only edema but also edema-induced gestational hypertension. On the other hand, since the increase in ICW shows a positive correlation with muscle mass [42], as a result of checking the SMM in this study, it was found that all groups significantly increased muscle mass after exercise compared to before (CON group: SMM: $z = -2.371$, $p = 0.018$, PE group: SMM: $z = -2.670$, $p = 0.008$; Table 2), which is thought to be due to the increase in SMM as part of the weight gain. In the difference in the amount of change in SMM, the PE group showed a significant increase compared to the CON group, and the results were consistent with previous studies that reported an increase in skeletal muscle after Pilates exercise [43,44]. In addition, the same result was obtained by a study showing that the group who did Pilates exercise for 8 weeks for pregnant women increased grip strength [45] and that symptoms of urinary incontinence, which are common in pregnant women, were more effectively reduced in the Pilates exercise group than in Kegel exercise one [46].

It is believed that static and dynamic Pilates movements of contraction and relaxation induce skeletal muscle increase, and this is thought to be a result partially supporting the significantly increased ICW level in the PE group presented above. Many previous studies reported that Pilates exercise reduced body fat and had a positive effect on blood lipid metabolism [25,26]. The increased body fat and blood lipid index during pregnancy may be partially reduced through Pilates exercise. In a study related to lipid indicators in pregnant women, Elena Rita [45] suggested that participation in Pilates exercise by overweight and obese pregnant women reduces the risk of diabetes and requires aerobic exercise to prevent weight gain. However, in this study, there was no significant difference in the amount of change (post − pre) between the PE group and the CON group. These results suggest that the subjects of this study were not overweight or obese pregnant women who could benefit from exercise intervention via an advantage in blood index [47] and that there were few statistically significant changes, including those with normal BMI levels. In addition, some studies examining the effect of exercise participation on the prevention of gestational diabetes regardless of obesity have shown contradictory results. The different results of several clinical studies related to pregnant women are probably due to the fact that the subjects participating in this study are pregnant women who have no choice but to show an imbalance in lipid metabolism, which is highly related to increased body fat, unlike the case in the general population. Therefore, in future studies, there seems to be a need for a more randomized trial on the type, intensity, and duration of exercise in preventing GDM (Gestational Diabetes Mellitus) [48].

### 4.2. Effect of Pilates on the Pelvic Stabilization Muscle Strength

During pregnancy, the anterior tilt of the pelvis disrupts the alignment of the spine and destabilizes the surrounding muscles, resulting in low back pain and pelvic pain [49,50]. Therefore, in order to alleviate the low back pain that occurs in most pregnant women, it is necessary to strengthen the muscles of the back and stabilize the hip joint [25]. Since Pilates exercise is known to induce movement

of the core muscles around the waist to improve the function of the muscles related to low back pain [51], it can be predicted that there will be a difference in the amount of change in HF, HA, and HS before and after. According to a previous study, in a randomized clinical trial in pregnant women, after 8 weeks of Pilates exercise, low back pain in pregnant women was significantly reduced, grip strength and lower limb flexibility increased, and spinal curvature was significantly improved [52]. As a result of confirming the difference in the amount of change in HF, HA, and HS before and after exercise in this study, it was found that it was significantly increased in the PE group compared to that in the CON group. These results suggest that the Pilates exercise program consists of movements that stimulate the core muscles related to pelvic stabilization, and as a result, muscle strength is thought to have increased. The results were similar to those of previous studies that reported improvement of hip function and muscle strength after Pilates exercise [53,54]. It is worth noting that the ASLR test was used for pregnant women. It reported a very high correlation with back pain, pelvic pain, and muscle function of load transfer over the pelvic reaction in pregnant women, presented as indicators of indirect confirmation of pain in pregnant women. It has also been reported that the decrease in ASLR is related to the deterioration of pelvic basal muscle function, which is important for childbirth and postpartum recovery [55].

Thus, Pilates is thought to be effective in decreasing or relieving low back pain commonly observed in pregnant women, as it increases the strength of muscles related to pelvic stabilization. However, some previous studies have reported no changes in core muscle strength in pregnant women after Pilates exercise. As mentioned earlier, the exacts effects would differ depending on the type, intensity, and duration of Pilates exercise; therefore, more studies are required to confirm the exact efficacy of Pilates.

### 4.3. Muscle Damage and Stress Markers

Since an increase in abdominal load due to fetal development causes pelvic pain, muscle stiffness, and a vicious cycle of body imbalance, inflammation and muscle damage are caused, and various methods have been used to reduce inflammation and muscle damage [56,57]. In particular, Pilates exercise is used as rehabilitation to treat or prevent inflammation and muscle damage by repeating muscle contraction and relaxation [56]. However, rather than the general public, the subjects who participated in this study were all pregnant women, and accurate information on the intensity of Pilates exercise is not yet known. Therefore, although in this study the activity of muscles related to body water balance and pelvic stabilization was shown by setting the intensity of Pilates exercise based on ACOG guidelines, the Pilates exercise program and intensity may prevent muscle damage and increased stress due to excessive and repetitive movements. Therefore, the muscle damage index and the stress index confirmed the Pilates efficiency applied in this study.

CK is a biochemical indicator of muscle damage that is present in muscle in large quantities and has a high correlation with the type of exercise and intensity, time, and quantity [32,33]. Furthermore, LDH, a representative muscle damage marker, is a result of muscle damage due to excessive exercise. It is known to increase with exercise intensity [58–60]. Interestingly, in the results of this study, CK and LDH between the periods within the group showed a tendency to increase significantly after the period compared to before (CK: $z = -1.700$, $p = 0.089$, $r = 0.425$; LDH: $z = -2.603$, $p = 0.009$, $r = 0.651$). AST, another indicator of muscle damage, is generally used as an indicator of liver damage, but is known to increase after muscle injury according to some studies [59,60]. As a result of confirming the difference in the amount of AST change in this study, similar to LDH, it was found to be significantly reduced in the PE group compared to in the CON group (Table 4), and in the difference between the periods within the group, only the CON group was significant compared with the before and after. It showed a tendency to increase. Previous studies have shown that these results may differ depending on the type of exercise, the degree of muscle contraction, and physical overload in the muscle damage of the human body [60], whereas these results show that the muscle damage

that occurs during pregnancy is partially reduced by Pilates. It is believed that the Pilates program conducted in this study improved muscle strength within the range that did not cause muscle damage.

Changes in body shape during pregnancy and pain such as low back pain induce stress in the tissues of cells, increasing the level of stress hormones along with inflammation [61]. In this study, as a result of confirming the difference in the amount of change in CRP, which is an inflammatory indicator that increases due to non-specific stress in the human body, there was no significant difference between the two groups. The CON group showed an increasing tendency, but there was no significant difference (Table 4). In addition, the stress hormone cortisol also increased significantly after in both groups compared to before (CON group: $z = -2.201$, $p = 0.028$; PE group: $z = -2.547$, $p = 0.011$; Figure 2); there was no significant difference in the amount of change between the two groups. This is believed to be due to the inevitable physiological changes that occur during pregnancy, and since there are various factors that cause inflammation and stress, further research is necessary. Furthermore, the small number of subjects who participated in this study seem to be insufficient to generalize the results. Recently, the fertility rate in South Korea has continued to decline, and there has been great difficulty in securing subjects to participate in studies. If more subjects are selected and studied in the future, it is expected that the effect of Pilates could be confirmed more concretely and scientifically.

## 5. Conclusions

This study confirmed the effects of 12 weeks of Pilates exercise on body composition, lipid metabolism, and pelvic stabilization-related muscle and muscle damage in pregnant women. Pilates exercise has been shown to improve body water balance and increase skeletal muscle. In addition, HF, HA, and HS, which affect muscles related to low back pain and pelvic stabilization, were improved through Pilates, while the index of muscle damage that could be increased by pregnancy was decreased. Therefore, Pilates exercise at the intensity conducted in this study is considered to be an effective and safe exercise that can strengthen the muscles related to pelvic stabilization within a range that does not cause muscle damage and stress in pregnant women.

## 6. Ethics Approval and Informed Consent

The research protocol was approved by the Bioethics Committee of Korea National Sports University (1263-201803-BR-001-01).

**Author Contributions:** Conceptualization, A.-H.H.; data curation, A.-H.H.; formal analysis, A.-H.H.; writing—original draft, A.-H.H.; project administration, Y.-J.J.; supervision, Y.-J.J.; validation, Y.-J.J.; writing—review and editing, Y.-J.J. All authors acknowledged responsibility for the full content of the submitted manuscript and approved their submission. All authors have read and agreed to the published version of the manuscript.

**Funding:** This research received no external funding.

**Conflicts of Interest:** The authors declare no conflict of interest.

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
