# Peer review of "Effect of Mat Pilates on Body Fluid Composition, Pelvic Stabilization, and Muscle Damage during Pregnancy"

_applsci, doi:10.3390/app10249111_

Round 1

Reviewer 1 Report

The authors raise extremely important research issues. The manuscript reads smoothly and is easy to understand.  The aims, scope, and results of the study are clearly stated.  I have very much enjoyed reading this paper. I find it interesting and clearly written, and satisfying also all the other publication criteria of the journal Applied Sciences.

  • Nevertheless, please underline in the introduction why pilates exercises were chosen from among many other methods.
  • Did freezing the samples at minus 80 degrees Celsius affect the test results? What does the literature describe about it?
  • Do the authors consider taking breathing techniques while exercising?
  • Please supplement the literature with more recent articles.
  • Please underline what is new at work compared to other publications.
  • In the discussion, please refer your results to other studies described in the literature.
  • In the future, please consider researching a larger research group.

Author Response

   " Please see the attachment "

Dear Reviewer:

I submitted a paper titled "The Effects of Matt Pilates on Fluid Composition, Pelvic Stabilization, and Muscle Damage During Pregnancy." Thesis was written by A-Hyun Hyun (Affiliation: Department of Exercise Physiology, Korea Physical Education University, e-mail: [email protected]) and Yoo-Jeong Jeon (Affiliation: Department of Sports & Health Science, Hanbat National University, e-mail: [email protected]) Co-authored. Thank you for your interest in this article.

The purpose of this study was to analyze the effects of mat pilates exercise performed at the intensity suggested by the U.S. College of Obstetrics and Gynecology on body composition, lipid profile, pelvic stabilization-related muscles, and muscle damage to confirm the effectiveness and appropriate intensity of Pilates exercise.

We believe that our study may provide basic data in the literature because studies demonstrating the efficacy of Pilates exercises with different types, intensity and frequency of exercise during pregnancy are insufficient. In particular, there is a lack of research on Korean pregnant women.

In addition, we believe that this paper will be interesting as our new research provides freshness and basic data for future health care.

This manuscript has not been published or published elsewhere, in part or in whole. All authors have approved the manuscript to be viewed and submitted to the journal. All study participants provided informed consent and the study design was approved by the appropriate ethics review committee.

Thank you for your consideration. We look forward to hearing from our dear reviewers.

Sincerely,

[Author’s name] Yoo-Jeong Jeon

-----------------------------------------------------------------------------------------

<Author's answer to the revision order-Reviewer 1>

  1. Nevertheless, please underline in the introduction why pilates exercises were chosen from among many other methods.

(lines 69~82)

Through recent research, it has been reported that Pilates exercise is suitable for pregnant women because it is composed of movements that can be safely performed indoors and stimulates the core muscles and pelvic stabilization muscles. In this regard, ACOG strongly recommends Pilates exercise for pregnant women. Are doing [25, 30]. Looking at the previous studies that suggested the effect of Pilates exercise for pregnant women, regular Pilates exercise reduces low back pain by enhancing respiratory muscle, flexibility, muscle strength, and postural stability, and improves basic physical strength such as cardiopulmonary function through whole body exercise. It has been reported to be effective [25, 26]. In addition, resistance exercise using props is known to have a positive effect on energy metabolism by increasing lean mass and basal metabolism. Particularly, maternity muscle exercises for pregnant women are effective in preventing back birth, and by strengthening the muscle strength of the abdomen and hip joints, the balance ability is improved to prevent falls and reduce neck pain, pelvic pain, and fatigue at the end of pregnancy [25]. Carolina et al. (2019) reported that pilates exercise improves postpartum depression by promoting the secretion of cranial nerve substances and dopamine [12].

  1. Did freezing the samples at minus 80 degrees Celsius affect the test results? What does the literature describe about it?

(lines 165~168)

We referred to previous research data, I think it is a reasonable method to prevent denaturation and stability of all analytes stored in deep freezer at –80 °C.

The content of the paper has been revised.

(Reference study)

  1. Yoshihisa, S.; Kiyoshi, I. Elucidation of stability profiles of common chemistry analytes in serum stored at six graded temperatures. Clin Chem Lab Med. 2019 Aug 27;57(9):1388-1396.
  2. Haslacher H, Szekeres T, Gerner M, Ponweiser E, Repl M, Wagner OF, et al. The effect of storage temperature fluctuations on the stability of biochemical analytes in blood serum. Clin Chem Lab Med 2017; 55:974–83.
  3. Ikeda K, Ichihara K, Hashiguchi T, Hidaka Y, Kang D, Maekawa M, et al. Evaluation of the short-term stability of specimens for clinical laboratory testing. Biopreserv Biobank 2015; 13:135–43.

  1. Do the authors consider taking breathing techniques while exercising?

(lines 52~54)

Related contents and references are attached as follows.

Through Pilates exercise, it stimulates the trunk muscles through chest eating and abdominal breathing, strengthens the deep muscles, and relieves psychological stability

(Dilek et al., 2018; Barakat et al., 2015).

  1. Please supplement the literature with more recent articles.

(lines 360~174), (437~446), (451~454)

We have updated to the latest information as follows. (Attachment of literature for 2018-2020)

(Attachment file number)

  1. Liyuan Z & Xinhua X (2018). The role of gut microbiota in the effects of maternal obesity during pregnancy on offspring metabolism. Bioscience Reports. 27:38(2)
  2. Mazzarino, M., D. Kerr, and M.E. Morris, Pilates program design and health benefits for pregnant women: A practitioners' survey. Journal of bodywork and movement therapies, 2018. 22(2): p. 411-417.
  3. Bahareh M, D., et al., [2020]. Designing an intervention program over the effects of Pilates on pregnancy outcomes among the pregnant women : A protocol study. International Journal of Surgery Protocols Volume 24, Pages27-30.
  4. Rita Santos-Rocha, Marco Branco, Liliana Aguiar, Filomena Vieira, António Prieto Veloso [2019]. Biomechanical Adaptations of Gait in Pregnancy: Implications for Physical Activity and Exercise. Exercise and Sporting Activity During Pregnancy. pp 95-134.
  5. Dilek S & Güder Y. (2018). The effect of pregnancy Pilates-assisted childbirth preparation training on childbirth fear and neonatal outcomes. Quality & Quantity. 52(6):2667–2679.
  6. Carolina C., Marlos D., Alan S. (2019). Efficacy of Regular Exercise During Pregnancy on the Prevention of Postpartum Depression. JAMA Netw Open. 2(1):186861.
  7. Hyun H & Yong C. (2019). Effects of 12-weeks pilates mat exercise on body composition, delivery confidence, and neck disability index in pregnant women. Sport Science. 36;2:43-55.
  8. Elena R, et al., Exercise during pregnancy and risk of preterm birth in overweight and obese women: a systematic review and meta-analysis of randomized controlled trials. Acta Obstet Gynecol Scand, 2017, 96(3):263-273.
  9. Pavithralochani V., et al., Efficacy of Kegel's Exercise vs Pilates in Subject with Urinary Incontinence during Pregnancy. Research Journal of Pharmacy and Technology. 2019, Volume : 12, Issue : 12
  10. Mei-Chen D., et al., Effects of physical exercise during pregnancy on maternal and infant in overweight and obese pregnant women: A meta-analysis. Birth, 2019, 46(2):211-221.
  11. Fatemeh N., et al., The effect of exercise on the prevention of gestational diabetes in obese and overweight pregnant women: a systematic review and meta-analysis. Diabetology & Metabolic Syndrome, 2019, volume 11, 72.
  12. Luciano R., et al., Effectiveness of a physical activity programme based on the Pilates method in pregnancy and labour. Enferm Clin, 2017;27(5):271-277
  13. Ika Oktaviani. Pilates workouts can reduce pain in pregnant women. Complement Ther Clin Pract, 2018, 31:349-351.

  1. Please underline what is new at work compared to other publications.

(lines 268~275)

We added the contents as follows. The reflected content is indicated by underlined below.

It is noteworthy that the ASLR test of was used for pregnant women. ASLR reported very high correlation with back pain, pelvic pain, and muscle function of load transfer over the pelvic reaction in pregnant women, presented as indicators of indirect confirmation of pain in pregnant women [31]. It has also been reported that the decrease in ASLR is related to the deterioration of pelvic basal muscle function, which is important for childbirth and postpartum recovery [50].

Thus, Pilates is thought to be effective in decreasing or relieving lower back pain commonly observed in pregnant women as it increases the strength of muscles related to pelvic stabilization.

(lines 224~227)

As a result of checking SMM in this study, it was found that all groups significantly increased muscle mass after death compared to before (CON group: SMM: z=-2.371, p=0.018, PE group: SMM: z=-2.670, p=0.008, Table. 2), which is thought to be due to the increase in SMM in part of the weight gain.

  1. In the discussion, please refer your results to other studies described in the literature.

Author's answer:

The discussion section was further explained as follows. Updated with the latest information along with supplementary literature. It is related to pregnant women and Pilates and has been modified as follows.

(lines 223~252)

4.1. Effect of Pilates on the Body compositions and lipid profiles

On the other hand, since the increase in ICW shows a positive correlation with muscle mass [37], as a result of checking SMM in this study, it was found that all groups significantly increased muscle mass after death compared to before (CON group: SMM: z=-2.371, p=0.018, PE group: SMM: z=-2.670, p=0.008, Table. 2), which is thought to be due to the increase in SMM in part of the weight gain. In the difference in the amount of change in SMM, the PE group showed a significant increase compared to the CON group, and the results were consistent with previous studies that reported an increase in skeletal muscle after Pilates exercise [38, 39]. In addition, the same result was as a result of a study showing that the group who did Pilates exercise for 8 weeks for pregnant women increased grip strength [40], and that the symptoms of urinary incontinence, which are common in pregnant women, were more effective in the Pilates exercise group than in Kegel exercise [41]. It is believed that static and dynamic pilates movements of contraction and relaxation induce skeletal muscle increase, and this is thought to be a result of partially supporting the significantly increased ICW level in the PE group presented above. Many previous studies that reported that pilates exercise reduced body fat and had a positive effect on blood lipid metabolism were reported [25, 26]. The increased body fat and blood lipid index during pregnancy may be partially reduced through pilates exercise. You can think of the possibility that there is. In a study related to lipid indicators in pregnant women, Elena Rita (40) suggested that participation in pilates exercise among overweight and obese pregnant women reduces the risk of diabetes and requires aerobic exercise to prevent weight gain. However, in this study, there was no significant difference in the amount of change (post-pre) between the PE group and the CON group. These results suggest that the subjects of this study were not overweight or obese pregnant women who could benefit from exercise intervention as an advantage in blood index [42], and that there were few statistically significant changes, including those with normal BMI levels. In addition, some studies examining the effect of exercise participation on the prevention of gestational diabetes regardless of obesity have shown contradictory results. The different results of several clinical studies related to pregnant women are probably due to the fact that the subjects participating in this study are pregnant women who have no choice but to show an imbalance in lipid metabolism, which is highly related to increased body fat, rather than the general population. Therefore, in future studies, there seems to be a need for a more randomized trial on the type, intensity, and duration of exercise in preventing GDM [43].

---------------------------------------------------------------------------

(lines 257~268)

4.2. Effect of Pilates on the pelvic stabilization muscle strength

Interestingly, since Pilates exercise is known to induce movement of the core muscles around the waist to improve the function of the muscles related to low back pain [46], it can be predicted that there will be a difference in the amount of change in HF, HA, and HS before and after. According to the preceding study, in a randomized clinical trial in pregnant women, after 8 weeks of Pilates exercise, lower back pain in pregnant women was significantly reduced [47], increased grip strength and lower limb flexibility, and spinal curvature was significantly improved [46]. As a result of confirming the difference in the amount of change in HF, HA, and HS before and after exercise in this study, it was found that it was significantly increased in the PE group compared to the CON group. These results suggest that the Pilates exercise program consists of movements that stimulate the core muscles related to pelvic stabilization, and as a result, muscle strength is thought to have increased. The results were similar to previous studies that reported improvement of hip function and muscle strength after Pilates exercise. [48, 49].

We have updated to the latest information as follows. (Attachment of literature for 2018-2020)

(Attachment file number)

  1. Liyuan Z & Xinhua X (2018). The role of gut microbiota in the effects of maternal obesity during pregnancy on offspring metabolism. Bioscience Reports. 27:38(2)
  2. Mazzarino, M., D. Kerr, and M.E. Morris, Pilates program design and health benefits for pregnant women: A practitioners' survey. Journal of bodywork and movement therapies, 2018. 22(2): p. 411-417.
  3. Bahareh M, D., et al., [2020]. Designing an intervention program over the effects of Pilates on pregnancy outcomes among the pregnant women : A protocol study. International Journal of Surgery Protocols Volume 24, Pages27-30.
  4. Rita Santos-Rocha, Marco Branco, Liliana Aguiar, Filomena Vieira, António Prieto Veloso [2019]. Biomechanical Adaptations of Gait in Pregnancy: Implications for Physical Activity and Exercise. Exercise and Sporting Activity During Pregnancy. pp 95-134.
  5. Dilek S & Güder Y. (2018). The effect of pregnancy Pilates-assisted childbirth preparation training on childbirth fear and neonatal outcomes. Quality & Quantity. 52(6):2667–2679.
  6. Carolina C., Marlos D., Alan S. (2019). Efficacy of Regular Exercise During Pregnancy on the Prevention of Postpartum Depression. JAMA Netw Open. 2(1):186861.
  7. Hyun H & Yong C. (2019). Effects of 12-weeks pilates mat exercise on body composition, delivery confidence, and neck disability index in pregnant women. Sport Science. 36;2:43-55.
  8. Elena R, et al., Exercise during pregnancy and risk of preterm birth in overweight and obese women: a systematic review and meta-analysis of randomized controlled trials. Acta Obstet Gynecol Scand, 2017, 96(3):263-273.
  9. Pavithralochani V., et al., Efficacy of Kegel's Exercise vs Pilates in Subject with Urinary Incontinence during Pregnancy. Research Journal of Pharmacy and Technology. 2019, Volume : 12, Issue : 12
  10. Mei-Chen D., et al., Effects of physical exercise during pregnancy on maternal and infant in overweight and obese pregnant women: A meta-analysis. Birth, 2019, 46(2):211-221.
  11. Fatemeh N., et al., The effect of exercise on the prevention of gestational diabetes in obese and overweight pregnant women: a systematic review and meta-analysis. Diabetology & Metabolic Syndrome, 2019, volume 11, 72.
  12. Luciano R., et al., Effectiveness of a physical activity programme based on the Pilates method in pregnancy and labour. Enferm Clin, 2017;27(5):271-277
  13. Ika Oktaviani. Pilates workouts can reduce pain in pregnant women. Complement Ther Clin Pract, 2018, 31:349-351.

  1. In the future, please consider researching a larger research group.

Author's answer:

For the following reasons, it was very difficult to recruit samples compared to the general public.

-As subjects for clinical research, pregnant women belong to the high-risk group and have difficulties in recruiting with voluntary consent.

-South Korea the fertility rate continues to decline, making it difficult to secure the sample size.

-It is difficult to obtain consent from related industries (obstetrics and gynecology hospitals, sports cultural facilities)

-Korean pregnant women have lived in a culture that reduces physical activity when they are pregnant, so the rate of participation in exercise is lower than in other countries.

In the future, we will conduct research with a little more sample size.

Reviewer 2 Report

There are a few suggestions to the authors:

  1. This study was a human study of pregnant women and should be approved by the IRB, but is not mentioned in the text. If approved by the IRB, please write it clearly.
  2. Are the participants randomly assigned to Pilates exercise or control group? Please explain clearly.
  3. How to decide the sample size was 30? Please explain clearly.
  4. The withdraw rate was too high, if this study was a RCT, please provide the flow diagram, and address the withdraw reasons of the participants.
  5. Since the withdraw rate is too high, it is recommended to use the intention-to-treat analysis method to avoid overestimating statistical results.
  6. Failure to take into account the differences between the two groups of basic information (pre-test) is also one of the factors to be considered to be comparison.

Author Response

Dear Reviewer:

I submitted a paper titled "The Effects of Matt Pilates on Fluid Composition, Pelvic Stabilization, and Muscle Damage During Pregnancy. Thesis was written by A-Hyun Hyun (Affiliation: Department of Exercise Physiology, Korea Physical Education University, e-mail: [email protected]) and Yoo-Jeong Jeon (Affiliation: Department of Sports & Health Science, Hanbat National University, e-mail: [email protected]) Co-authored. Thank you for your interest in this article.

The purpose of this study was to analyze the effects of mat pilates exercise performed at the intensity suggested by the U.S. College of Obstetrics and Gynecology on body composition, lipid profile, pelvic stabilization-related muscles, and muscle damage to confirm the effectiveness and appropriate intensity of Pilates exercise.

We believe that our study may provide basic data in the literature because studies demonstrating the efficacy of Pilates exercises with different types, intensity and frequency of exercise during pregnancy are insufficient. In particular, there is a lack of research on Korean pregnant women.

In addition, we believe that this paper will be interesting as our new research provides freshness and basic data for future health care.

This manuscript has not been published or published elsewhere, in part or in whole. All authors have approved the manuscript to be viewed and submitted to the journal. All study participants provided informed consent and the study design was approved by the appropriate ethics review committee.

Thank you for your consideration. We look forward to hearing from our dear reviewers.

Sincerely,

[Author’s name] Yoo-Jeong Jeon

-----------------------------------------------------------------------------------------

<Author's answer to the revision order-Reviewer 2>

  1. This study was a human study of pregnant women and should be approved by the IRB, but is not mentioned in the text. If approved by the IRB, please write it clearly.

Author's answer:

The research protocol was approved by the Bioethics Committee of Korea National Sports University (1263-201803-BR-001-01).

- Attachment of ethical approval result notice, Also attached to the thesis content.

(Please see the attachment)

-Information of the person in charge of bioethics is as follows.

Name of person in charge: Kim Ji-yeon

Korea Sports University Industry-Academic Cooperation Foundation

1239 Yangjae-daero, Songpa-gu, Seoul

T: 02-410-6724 / F: 02-410-6729

E: [email protected]

  1. Are the participants randomly assigned to Pilates exercise or control group? Please explain clearly.

It has been modified as follows.

This study is a voluntary participation study (Voluntary Participation Study).

Recruitment of the candidates was announced through the Cultural Center of C Women's Hospital located in Bundang, and pregnant women were recruited with voluntary consent.

In the data collected in this study, 16 people were finally analyzed. (Pilates exercise group (9 people) and control group (7 people) were analyzed as the study results).

The reason for dropouts was that they requested to stop exercising due to physical or medical problems or personal reasons, or because pregnant women with low attendance rates during the study period were excluded from statistics.

  1. How to decide the sample size was 30? Please explain clearly.

-As subjects for clinical research, pregnant women belong to the high-risk group and have difficulties in recruiting with voluntary consent.

-As the fertility rate in Korea continues to decline, I wanted to increase the number of samples, but there is a great difficulty in securing it.

-It is also difficult to obtain consent from related industries (obstetrics and gynecology hospitals, sports cultural facilities).

-Korean pregnant women have lived in a culture that reduces physical activity when they are pregnant, so their participation in exercise is lower than in other countries.

For the above reasons, it was very difficult to recruit samples compared to the general public.

In the future, we will conduct research with a little more sample size.

  1. The withdraw rate was too high, if this study was a RCT, please provide the flow diagram, and address the withdraw reasons of the participants.

This study is not an RCT study. This is a voluntary participation study of the subject of the study. (Voluntary Participation Study)

Dropouts occurred for the following reasons.

-The personal condition of pregnant women is very diverse. There are also small symptoms such as a cold or fatigue, and exercise loss occurred due to medical problems such as disk problems and autophagy.

-Among women in late pregnancy, there was a case of dropping out due to early delivery.

-Pregnant women with a low attendance rate were excluded from the statistics to verify the exercise effect.

  1. Since withdraw rate is too high, it is recommended to use the intention-to-treat analysis method to avoid overestimating statistical results.

Thank you very much for the good advice.

Future research will use Intention-to-treat analysis.

(The reason why Intention-to-treat analysis was not used in this study)

The treatment intention analysis, which includes data of dropouts in the results, takes a statistically conservative position on the effectiveness of interventions.

This is because if the data of the dropouts or the participants of the experiment who are not sincere are applied to the results, the data of those who did not faithfully undergo the intervention are included and the effectiveness of the intervention is not sufficiently reflected, and the effect may be weakly measured.

(Alshurafa et al., 2012; Batistaet al., 2019)

(Reference study)

  1. Alshurafa, M., Briel, M., Akl, E. A., Haines, T., Moayyedi, P., Gentles, S. J., ... Lamontagne, F. (2012). Inconsistent definitions for intention-to-treat in relation to missing outcome data: Systematic review of the methods literature. PLoS One, 7(11). doi:10.1371/journal.pone.0049163.
  2. Batista, K. B. D. S. L., Thiruvenkatachari, B., & O'Brien, K. (2019). Intention-to-treat analysis: Are we managing dropouts and missing data properly in research on orthodontic treatment? A systematic review. American Journal of Orthodontics and Dentofacial Orthopedics, 155(1), 19-27.

doi:10.1016/j.ajodo.2018.08.013

  1. Failure to take into account the differences between the two groups of basic information (pre-test) is also one of the factors to be considered to be comparison.

The reasons for dropout of the control group (non-exercise) among the two groups are as follows. They were unable to participate in the post-test for the following reasons.

-Non-participation due to parenting

-Non-participation of pregnant women who work

-Non-participation due to personal reasons (family related reasons)

Reviewer 3 Report

General comments

This is an interesting study that would benefit from a stronger rationale for the muscle damage and body composition analysis used. The study design needs great explanation, as it isn’t clear whether it was a randomised controlled trial as no information on group allocation, or blinding is provided. The statistical methods, although adequate, could be improved by using effect sizes and interpreting their magnitudes using a scale such as that of Cohen.

Specific comments

Introduction

  • The results reported in the abstract do not include any data or effect sizes…

Introduction

  • The introduction is generally well-written and outlines a clear rationale for the study. However, the rationale for the tests for muscle damage and body composition needs to be clearer. For muscle damage, the possibility that Pilates could cause muscle damage is introduced (line 58), but with no justification or supporting references. Likewise, body composition appears for the first time (line 63), again with no justification of references. What evidence is there that body composition and muscle damage are worthy of study?
  • A reference is needed to support the statement in lines 48-50.

Methods

  • How were the participants allocated to groups? Please provide details of the randomisation process if one was used. If it wasn’t, this needs to be made clear along with a justification for doing so.
  • Were the assessors blinded to the group allocation?

Results

  • A comparison of baseline values between groups is required.
  • When reporting Mann-Whitney test statistics, if you divide the Z value by the square root of N, this provides an effect size measure with a distribution that approximates a correlation coefficient so a scale of magnitudes such as that of Cohen can be used. This should be done throughout the results section.
  • Figure 1 and Figure 2 would be easier to interpret without the individual results that mask the error bars on the figures.
  • The methods did not refer to back-pain related strength for the de Groot test, whereas this is introduced in the results in the title of Figure 1. This needs to be consistent throughout the manuscript, and if back pain-related is used, then the rationale for this terminology needs to be explained.

Discussion

  • Was the weeks of gestation at inclusion in this study similar to that of other studies?
  • There is no need to replicate results in the discussion, other than in qualitative terms unless results are being compared with other studies and effect sizes are used.
  • The reference for de Groot should be in square brackets and “de Groot” used, not “Groot”.
  • How do the results of the present study compare to those of previous research? Were the results of studies 36 and 37 similar to those of the current study? Was the increase the same, or did it differ potentially due to pregnancy?
  • If pain was an objective of the study, it should have been measured or at least the use of the de Groot ASLR test as a proxy mentioned in the methods.
  • The discussion around sample size and differences would be strengthened by reporting effect statistics. For instance, moderate effects that aren’t significant would indicate that the use of more subjects might have attained statistical significance, whereas a trivial (and non-significant) effect size would not support this assumption.

Author Response

Dear Reviewer:

I submitted a paper titled "The Effects of Matt Pilates on Fluid Composition, Pelvic Stabilization, and Muscle Damage During Pregnancy. Thesis was written by A-Hyun Hyun (Affiliation: Department of Exercise Physiology, Korea Physical Education University, e-mail: [email protected]) and Yoo-Jeong Jeon (Affiliation: Department of Sports & Health Science, Hanbat National University, e-mail: [email protected]) Co-authored. Thank you for your interest in this article.

The purpose of this study was to analyze the effects of mat pilates exercise performed at the intensity suggested by the U.S. College of Obstetrics and Gynecology on body composition, lipid profile, pelvic stabilization-related muscles, and muscle damage to confirm the effectiveness and appropriate intensity of Pilates exercise.

We believe that our study may provide basic data in the literature because studies demonstrating the efficacy of Pilates exercises with different types, intensity and frequency of exercise during pregnancy are insufficient. In particular, there is a lack of research on Korean pregnant women.

In addition, we believe that this paper will be interesting as our new research provides freshness and basic data for future health care.

This manuscript has not been published or published elsewhere, in part or in whole. All authors have approved the manuscript to be viewed and submitted to the journal. All study participants provided informed consent and the study design was approved by the appropriate ethics review committee.

Thank you for your consideration. We look forward to hearing from our dear reviewers.

Sincerely,

[Author’s name] Yoo-Jeong Jeon

----------------------------------------------------------------------------------------

< Author's answer to the revision order-Reviewer 3 >

General comments

  1. The study design needs great explanation, as it isn’t clear whether it was a randomised controlled trial as no information on group allocation, or blinding is provided. The statistical methods, although adequate, could be improved by using effect sizes and interpreting their magnitudes using a scale such as that of Cohen.

It has been modified as follows.

This study is a voluntary participation study (Voluntary Participation Study).

Recruitment of candidates was announced through the Cultural Center of Women's Hospital c, located in Gyeonggi-do, Korea, and a total of 30 pregnant women were recruited with voluntary consent.

For the following reasons, it was very difficult to recruit samples compared to the general public.

-As subjects for clinical research, pregnant women belong to the high-risk group and have difficulties in recruiting with voluntary consent.

-Korea's fertility rate continues to decline, making it difficult to secure a sample.

-It is difficult to obtain consent from related industries (obstetrics and gynecology hospitals, sports cultural facilities)

-Korean pregnant women have lived in a culture that reduces physical activity when they are pregnant, so the rate of participation in exercise is lower than in other countries.

In the future, we will conduct research with a little more sample size.

Specific comments

  1. The results reported in the abstract do not include any data or effect size.

(lines 22~34)

Author's answer: In the thesis, the content of Abstract was added and modified in more detail. 

  1. Introduction

The introduction is generally well-written and outlines a clear rationale for the study. However, the rationale for the tests for muscle damage and body composition needs to be clearer. For muscle damage, the possibility that Pilates could cause muscle damage is introduced (line 58), but with no justification or supporting references. Likewise, body composition appears for the first time (line 63), again with no justification of references. What evidence is there that body composition and muscle damage are worthy of study?

A reference is needed to support the statement in lines 48-50.

The discussion section was further explained as follows. It has been updated to the latest information along with the supplementary literature and has been modified as follows.

(lines 223~227), (236~252)

<Correction of evidence of body composition>

As a result of checking SMM in this study, it was found that all groups significantly increased muscle mass after death compared to before (CON group: SMM: z=-2.371, p=0.018, PE group: SMM: z=-2.670, p=0.008, Table. 2), which is thought to be due to the increase in SMM in part of the weight gain. In the difference in the amount of change in SMM, the PE group showed a significant increase compared to the CON group, and the results were consistent with previous studies that reported an increase in skeletal muscle after Pilates exercise [38, 39].

Many previous studies that reported that pilates exercise reduced body fat and had a positive effect on blood lipid metabolism were reported [25, 26]. The increased body fat and blood lipid index during pregnancy may be partially reduced through pilates exercise. You can think of the possibility that there is.

In a study related to lipid indicators in pregnant women, Elena Rita Magro Malosso et al. (2017) suggested that participation in Pilates exercise among overweight and obese pregnant women reduces the risk of diabetes and requires aerobic exercise to prevent weight gain.

However, in this study, there was no significant difference in the amount of change (post-pre) between the PE group and the CON group.

These results show that the subjects of this study were not overweight or obese pregnant women who could benefit from exercise intervention as an advantage in blood index [Mei-Chen Du, 2019], and pregnant women with normal BMI levels were included, so there was little statistically significant change. I think it was.

In addition, some studies examining the effect of exercise participation on the prevention of gestational diabetes regardless of obesity have shown contradictory results [Fatemeh Nasiri-Amiri, 2019].

The different results of several clinical studies related to pregnant women are probably due to the fact that the subjects participating in this study are pregnant women who have no choice but to show an imbalance in lipid metabolism, which is highly related to increased body fat, rather than the general population.

Therefore, in future studies, there seems to be a need for a more randomized test on the type, intensity, and duration of exercise in preventing GDM [Fatemeh Nasiri-Amiri, 2019].

  1. Hyun H & Yong C. (2019). Effects of 12-weeks pilates mat exercise on body composition, delivery confidence, and neck disability index in pregnant women. Sport Science. 36;2:43-55.
  2. Martin, A., et al., Pilates for Pregnant Women: A Healthy Alternative. J Women's Health Care, 2017. 6(366): p. 2167-0420.1000366.
  3. Elena R, et al., Exercise during pregnancy and risk of preterm birth in overweight and obese women: a systematic review and meta-analysis of randomized controlled trials. Acta Obstet Gynecol Scand, 2017, 96(3):263-273.
  4. Mei-Chen D., et al., Effects of physical exercise during pregnancy on maternal and infant in overweight and obese pregnant women: A meta-analysis. Birth, 2019, 46(2):211-221.
  5. Fatemeh N., et al., The effect of exercise on the prevention of gestational diabetes in obese and overweight pregnant women: a systematic review and meta-analysis. Diabetology & Metabolic Syndrome,2019,volume 11, 72.

--------------------------------------------------------------------------

The discussion section was further explained as follows. Currently, there is not much literature, so it was insufficient, but it was revised as follows with supplementation.

< Corrected evidence of muscle damage >

There is a result of previous studies that the damage to the muscle fibers of the human body may appear differently depending on the type of exercise, the degree of muscle contraction or physical overload (1),

CK is present in large amounts in muscles, fluctuates significantly with exercise, has a high correlation with exercise intensity, time, and quantity, and is used as a biochemical indicator of muscle damage (2.3).

However, there have been no studies on the effects of Pilates exercise by pregnant women on muscle damage.

As such, since there is no previous data conducted by Pilates exercise program for pregnant women, this study was also concerned about the consequences of muscle damage in pregnant women.

It is interpreted that the Pilates exercise program conducted in this study improved muscle strength within the range that did not cause muscle damage.

(Reference study)

(1) Falvo, M. J., & Bloomer, R. J. (2006) Review of exercise-induced muscle injury: Relevance for athletic populations. Res. Sports Med., 14(1): 65-82.

(2) Brancaccio P, Maffulli N, Limongelli FM. (2007). Creatine kinase monitoring in sport medicine. British Medical Bulletin, 81-82:209-230

(3) Clarkson PM, Hubal M.J. (2002) Exercise-Induced Muscle Damage in Humans. American Journal of Physical Medicine Rehabilitation, 81(Suppl):52-69.

  1. Methods

How were the participants allocated to groups? Please provide details of the randomisation process if one was used. If it wasn’t, this needs to be made clear along with a justification for doing so.

Were the assessors blinded to the group allocation?

It has been modified as follows with the same contents as the third answer.

This study is a voluntary participation study (Voluntary Participation Study).

A total of 30 pregnant women were recruited through the Cultural Center of the c Women's Hospital located in Bundang.

Participants were selected as a control group (15 people) and a Pilates group (15 people) who did not participate in the exercise, and were assigned to two groups.

In the data collected in this study, 16 people were analyzed except for 14 people, different from the number of people initially recruited.

The Pilates exercise group (9 subjects) and the control group (7 subjects) were analyzed as the study results.

The reasons for dropouts are as follows.

Pilates exercise group was dropped out for the following reasons.

-The personal condition of pregnant women is very diverse. There are also small symptoms such as a cold or fatigue, and exercise loss occurred due to medical problems such as disk problems and autophagy.

-Among women in late pregnancy, there was a case of dropping out due to early delivery.

-Pregnant women with a low attendance rate were excluded from the statistics to verify the exercise effect.

Among the two groups, the reasons for dropout of the control group (non-exercise) are as follows. They were unable to participate in the post-test for the following reasons.

-Non-participation due to parenting

-Non-participation of pregnant women who work

-Non-participation due to personal reasons (family related reasons)

---------------------------------------------------------------------------

  1. Results
  2. A comparison of baseline values between groups is required.

When reporting Mann-Whitney test statistics, if you divide the Z value by the square root of N, this provides an effect size measure with a distribution that approximates a correlation coefficient so a scale of magnitudes such as that of Cohen can be used. This should be done throughout the results section.

I am new to thesis work experience, so I am lacking, but I will better design and prepare for future research. Thank you very much for your interest in this paper and for your good advice.

  1. Figure 1 and Figure 2 would be easier to interpret without the individual results that mask the error bars on the figures.

(lines 7 of 14), (lines 8 of 14)

In the thesis, the result of covering the error bars of the figure has been modified to make it easier to interpret (Figure 1, Figure 2).

  1. The methods did not refer to back-pain related strength for the de Groot test, whereas this is introduced in the results in the title of Figure 1.

This needs to be consistent throughout the manuscript, and if back pain-related is used, then the rationale for this terminology needs to be explained.

The thesis has been revised well. (lines 7 of 14)

Incorrect content in the thesis and incorrect parts of the title in Figure 1. have also been corrected. (For example: back pain related → pelvic stabilization)

Figure 1. Effect of Pilates exercise on pelvic stabilization muscle strength

------------------------------------------------------------------

  1. Discussion

  1. Was the weeks of gestation at inclusion in this study similar to that of other studies?

In this study, the gestation period was 16-24 weeks, similar to the previous studies.

(Reference study)

  1. Between 16 and 17 weeks of gestation (Mette G Backhausen et al, PLoS One. 2017 Sep 6;12(9)
  2. 14-20 weeks gestation(Araceli Navas et al, BMC Pregnancy Childbirth. 2018 Apr 11;18(1):94)
  3. gestational week < 18 (Kirsti K Garnæs et al, BMC Pregnancy Childbirth. 2018 Jan 8;18(1):18)
  4. 8 weeks or more (Luciano R., et al, Enferm Clin, 2017;27(5):271-277)
  5. Previous studies have shown that physical activity during pregnancy is safe and desirable in the absence of obstetric or medical complications or contraindications, and pregnant women should be encouraged to continue or initiate safe physical activity.(Physical activity and exercise during pregnancy and postpartum period: ACOG Committee comment, number 804, Obstet Gynecol

. 2020 Apr;135(4):e178-e188.).

  1. There is no need to replicate results in the discussion, other than in qualitative terms unless results are being compared with other studies and effect sizes are used.

The reference for de Groot should be in square brackets and “de Groot” used, not “Groot”.

Author's answer: The thesis has been revised and supplemented.

(lines 135~137)

*Before = HP was measured using Groot's [24] active straight leg raising (ASLR) test; the participant was lying supine with both hands resting on the floor and the feet shoulder-width apart.

*After = HP was measured using de Groot active straight leg raise (ASLR) test. The participant was lying supine with both hands on the floor and feet shoulder-width apart [24].

  1. How do the results of the present study compare to those of previous research?

Were the results of studies 36 and 37 similar to those of the current study?

Was the increase the same, or did it differ potentially due to pregnancy?

It is explained in the thesis section as follows. A literature with similar results is attached.

(lines 223~252), (lines 257~268)

  1. Interestingly, since Pilates exercise is known to induce movement of the core muscles around the waist to improve the function of the muscles related to low back pain, it can be predicted that there will be a difference in the amount of change in HF, HA, and HS before and after [Luciano Rodríguez-Díaz, 2017].
  2. In a previous study of randomized clinical trials in pregnant women, after the Pilates exercise program, the pain of pregnant women was significantly reduced, the grip strength and lower limb flexibility increased, and the spinal curvature was remarkably improved [Ika Oktaviani 2018], [Luciano Rodríguez-Díaz, 2017].
  3. As a result of confirming the difference in the amount of change in HF, HA, and HS before and after exercise in this study, it was found that it was significantly increased in the PE group compared to the CON group. These results suggest that the Pilates exercise program consists of movements that stimulate the core muscles related to pelvic stabilization, and as a result, muscle strength is thought to have increased. The results were similar to previous studies that reported improvement of hip function and muscle strength after Pilates exercise.

(Melissa et al. 2018; Rodríguez et al. 2017).

  1. In the preceding study, the subjects completed 8 weeks of Pilates training and conducted a lumbar pelvic stability test.As the control group showed a large difference in grades in the test, regular Pilates exercise improved a lot in stabilization of the pelvis. Showed. In this study, Pilates improved the stability and flexibility of the pelvis. These results are the same as in previous studies (Harrington L. 2005, Davies R. Kish R. 1998, Segal A, Neil H, Basford R. 2004).

(Attachment file number)

  1. Liyuan Z & Xinhua X (2018). The role of gut microbiota in the effects of maternal obesity during pregnancy on offspring metabolism. Bioscience Reports. 27:38(2)
  2. Mazzarino, M., D. Kerr, and M.E. Morris, Pilates program design and health benefits for pregnant women: A practitioners' survey. Journal of bodywork and movement therapies, 2018. 22(2): p. 411-417.
  3. Bahareh Mothaghi Dastenaei,Fereshteh Aein,Faranak Safdari,Zohreh Karimiankakolak. [2020].DesigninganinterventionprogramovertheeffectsofPilatesonpregnancyoutcomesamongthepregnantwomen:Aprotocolstudy. International Journal of Surgery ProtocolsVolume 24,Pages27-30.
  4. Rita Santos-Rocha, Marco Branco, Liliana Aguiar, Filomena Vieira, António Prieto Veloso [2019]. Biomechanical Adaptations of Gait in Pregnancy: Implications for Physical Activity and Exercise. Exercise and Sporting Activity During Pregnancy. pp 95-134.
  5. Dilek S & Güder Y. (2018). The effect of pregnancy Pilates-assisted childbirth preparation training on childbirth fear and neonatal outcomes. Quality & Quantity. 52(6):2667–2679.
  6. Carolina C., Marlos D., Alan S. (2019). Efficacy of Regular Exercise During Pregnancy on the Prevention of Postpartum Depression. JAMA Netw Open. 2(1):186861.
  7. Hyun H & Yong C. (2019). Effects of 12-weeks pilates mat exercise on body composition, delivery confidence, and neck disability index in pregnant women. Sport Science. 36;2:43-55.
  8. Elena R, et al., Exercise during pregnancy and risk of preterm birth in overweight and obese women: a systematic review and meta-analysis of randomized controlled trials. Acta Obstet Gynecol Scand, 2017, 96(3):263-273.
  9. Pavithralochani V., et al., Efficacy of Kegel's Exercise vs Pilates in Subject with Urinary Incontinence during Pregnancy. Research Journal of Pharmacy and Technology. 2019, Volume : 12, Issue : 12
  10. Mei-Chen D., et al., Effects of physical exercise during pregnancy on maternal and infant in overweight and obese pregnant women: A meta-analysis. Birth, 2019, 46(2):211-221.
  11. Fatemeh N., et al., The effect of exercise on the prevention of gestational diabetes in obese and overweight pregnant women: a systematic review and meta-analysis. Diabetology & Metabolic Syndrome,2019,volume 11, 72.
  12. Luciano R., et al., Effectiveness of a physical activity programme based on the Pilates method in pregnancy and labour. Enferm Clin, 2017;27(5):271-277
  13. Ika Oktaviani.Pilates workouts can reduce pain in pregnant women. Complement Ther Clin Pract, 2018, 31:349-351.

  1. If pain was an objective of the study, it should have been measured or at least the use of the de Groot ASLR test as a proxy mentioned in the methods.

Author's answer: We added the contents as follows.

Note that in this study, HF, which was measured in pregnant women, was measured using the ASLR test, a method of de Groot. ASLR was reported to show a very high correlation with low back pain, pelvic pain, and muscle function of load transfer over the pelvic region of pregnant women, and was suggested as an index to indirectly identify the pain of pregnant women. In addition, it has been reported that a decrease in ASLR is associated with a decrease in pelvic basal muscle function, which is important for childbirth and postpartum recovery [38].

Therefore, Pilates exercise is considered to be an effective exercise method to reduce or alleviate low back pain that is common among pregnant women by increasing muscle strength related to hip joint stabilization.

  1. The discussion around sample size and differences would be strengthened by reporting effect statistics. For instance, moderate effects that aren’t significant would indicate that the use of more subjects might have attained statistical significance, whereas a trivial (and non-significant) effect size would not support this assumption.

Thank you very much for the good advice.

(The reason why Intention-to-treat analysis was not used in this study)

The treatment intention analysis, which includes data of dropouts in the results, takes a statistically conservative position on the effectiveness of interventions.

This is because if the data of the dropouts or the participants of the experiment who are not sincere are applied to the results, the data of those who did not faithfully undergo the intervention are included and the effectiveness of the intervention is not sufficiently reflected, and the effect may be weakly measured.

(Alshurafa et al., 2012; Batistaet al., 2019)

This study is a Voluntary Participation Study, and recruitment was announced through the Cultural Center of Women's Hospital c, located in Gyeonggi-do, South Korea, and was recruited with the voluntary consent of pregnant women.

As the fertility rate in Korea continued to decrease, it was very difficult to secure the sample number.Because pregnant women belong to high-risk groups, it was difficult to obtain consent from related industries (obstetrics and gynecology hospitals, sports cultural facilities). Because they came, the participation rate in the movement is low compared to other countries.

In the future, I will try to conduct research with a little more sample size. Thank you!!

(Reference study)

  1. Alshurafa, M., Briel, M., Akl, E. A., Haines, T., Moayyedi, P., Gentles, S. J., ... Lamontagne, F. (2012). Inconsistent definitions for intention-to-treat in relation to missing outcome data: Systematic review of the methods literature. PLoS One, 7(11). doi:10.1371/ journal. Pone .0049163.
  2. Batista, K. B. D. S. L., Thiruvenkatachari, B., & O'Brien, K. (2019). Intention-to-treat analysis: Are we managing dropouts and missing data properly in research on orthodontic treatment? A systematic review. American Journal of Orthodontics and Dentofacial Orthopedics, 155(1), 19-27.

doi:10.1016/ j. ajodo.2018.08.013

Round 2

Reviewer 2 Report

According to the author's reply, there are no other comments.

Author Response

The overall content and flow of the thesis have been revised.

(Revised part : Abstract, Introduction, Results, Discussion, References)

We have completed an English editing service that checks the improvement of grammer, spelling, and style of thesis.

" Please see the attachment "

Reviewer 3 Report

General comments

The authors have made some of the changes requested, however, several questions remain, particularly about study design. The authors were asked to explain the design in terms of blinding, group allocation, etc, but provided none of this information in their response. This needs to be answered before the paper could be considered for publication. In addition, it was hard to follow the changes made, as although some sections were highlighted in red, not all changes were marked. All changes must be marked with track changes, not just some of the changes made.

Specific comments

Abstract

  • The results reported in the abstract now include the Z score for the Mann-Whitney U test, however, these scores needed to be modified for interpretation as effect sizes, as described in the first review.

Introduction

  • Although the introduction was rewritten to provide additional justification, it is now extremely long and does not flow well. The introduction did not need to be completely rewritten it just needed some evidence for some of the hypotheses. There is still no justification for the muscle damage claims, with only one sentence seemingly added in lines 87-89 without a supporting reference.

Methods

  • No information has been provided in response to the questions about group allocation and randomisation. The response to this question was to say the study was a voluntary participation study, which is not an experimental design. In fact, all studies should be voluntary participation. Please provide the following information:
    • How were participants allocated to the experimental and control groups?
    • Were the assessors blinded to the group allocation?

Results

  • A comparison of baseline values between groups is still required.
  • The comment in response to the request to transform Z into r values that can be interpreted as effect size measures was answered but no changes were made. These changes need to be made in a revised submission. 

Author Response

Response to Reviewer 6 Comments

Point 1: General comments

The authors have made some of the changes requested, however, several questions remain, particularly about study design. The authors were asked to explain the design in terms of blinding, group allocation, etc, but provided none of this information in their response.

Response1: It has been modified as follows. (line 104~117)

This study was a cluster randomized controlled experiment. A total of 16 subjects recruited were Pilates exercise group (PE, n=9; age:31.78±4.68 years, body weight (BW):, 57.97±7.91 kg, height: 160.56±3.78 cm, using the block randomization method). body fat mass (BFM): 18.40±4.92 g, BMI: 39.57±4.52, skeletal muscle mass (SMM): 21.10±2.66 g) and control (CON, n=7, age:32.00±3.46 years, BW:, 53.67 ±5.70 kg, height: 161.29±3.54 cm, BFM: 15.86±3.06 g, BMI: 37.81±3.37, SMM: 20.09±2.00 g) were randomly assigned.

In addition, the comparison of the reference values between groups was performed by obtaining Cohen's r values. The research protocol was approved by the Bioethics Committee of Korea National Sports University (1263-201803-BR-001-01).

For the explanation of the above block randomization method, refer to the paper below.

https://doi.org/10.4097/kja.19049

Chi-Yeon Lim, Junyong In., Randomization in clinical studies. 2019, Korean Journal of Anesthesiology, 72(3):221-232.

------------------------------------------------------------------------

Point 2: This needs to be answered before the paper could be considered for publication. In addition, it was hard to follow the changes made, as although some sections were highlighted in red, not all changes were marked. All changes must be marked with track changes, not just some of the changes made.

Response 2: All changes have been “highlighted” as “Track Changes”.

-------------------------------------------------------------------------

Point 3: Specific comments

Abstract

The results reported in the abstract now include the Z score for the Mann-Whitney U test, however, these scores needed to be modified for interpretation as effect sizes, as described in the first review.

Response 3: (line 21~39), (line 186~192), (222~223)

We have completed the request for conversion of the paper, abstract, table and to r value.

(Table 2, Table 3, Table 4)

------------------------------------------------------------------------

Point 4: Introduction

Although the introduction was rewritten to provide additional justification, it is now extremely long and does not flow well. The introduction did not need to be completely rewritten it just needed some evidence for some of the hypotheses. There is still no justification for the muscle damage claims, with only one sentence seemingly added in lines 87-89 without a supporting reference.

Response 4: The contents have been revised by adding references to the introduction and discussion as follows.

<Revision contents of the introduction part> (line 83~97)

Particularly, since repetitive Pilates movements and excessive muscle use in pregnant women have the potential to cause more muscle damage as well as damage to joints [30], the Pilates exercise program for pregnant women should be carefully selected. CK is an index indicating the degree of muscle damage. The blood CK concentration of pregnant women before 34 weeks is correlated with gestational hypertension [31], and the level is high depending on the activity with high exercise intensity [32,33]. Previous studies have suggested that Pilates exercise using resistance tools directly stimulates the muscles to induce muscle damage and increases CK concentration [30,31], whereas the exercise group in the 12-week yoga exercise group compared unfavorably to the control group. It was suggested that the level of inflammation markers decreased, thereby reducing inflammation [34]. As such, the different results of the CK study are due to the fact that the results of muscle injury vary greatly depending on the intensity of the exercise. In particular, studies on Pilates exercise and CK studies in pregnant women are insufficient, and additional verification is needed. Therefore, it is necessary to first evaluate the exercise experience and physical ability of the pregnant woman before pregnancy, collect the mother’s information, and then divide and program the exercise intensity according to the pregnancy week.

(The added references are as follows.) (line 447~461)

  1. J, Kim., et al., The effects of pilates exercise on lipid metabolism and inflammatory cytokines mRNA expression in female undergraduates, Journal of Exercise Nutrition & Biochemistry, 2014;18(3):267-275.
  2. Horjus D., et al., Creatine kinase is associated with blood pressure during pregnancy, Journal of Hypertension, 2019. 37(7)p1467-1474.
  3. Eskild S., et al., Major increase in creatine kinase after intensive exercise, Tidsskr Nor Legeforen, 2019. Apr 25;139(8).
  4. Juviane M., et al.,Hemodynamic and creatine kinase changes after a 12-week equipment-based Pilates training program in hypertensive women, Journal of Bodywork and Movement Therapies, 2020. 24(4)496-502.
  5. Sung., H., et al., Effects of yoga exercise on maximum oxygen uptake, cortisol level, and creatine kinase myocardial bond activity in female patients with skeletal muscle pain syndrome, Journal of physical therapy science. 2015 May;27(5):1451-3

<Content of revision in discussion> (line 320~335)

4.3. muscle damage and stress maker

CK is a biochemical indicator of muscle damage and is present in muscle in large quantities, and has a high correlation with the type of exercise and intensity, time, and quantity [32,33]. Furthermore, LDH, a representative muscle damage marker, is a result of muscle damage due to excessive exercise. It is known to increase [58–60]. Interestingly, in the results of this study, CK and LDH between the periods within the group showed a tendency to increase significantly after the period compared to before (CK: z = −1.700, p = 0.089, r = 0.425, LDH: z = −2.603, p = 0.009, r = 0.651). AST, another indicator of muscle damage, is generally used as an indicator of liver damage, but is known to increase after muscle injury according to some studies [59,60]. As a result of confirming the difference in the amount of AST change in this study, similar to LDH, it was found to be significantly reduced in the PE group compared to the CON group (Table 4), and in the difference between the periods within the group, only the CON group was significantly compared with the before and after It showed a tendency to increase. Previous studies have shown that these results may differ depending on the type of exercise, the degree of muscle contraction, and physical overload in the muscle damage of the human body [60], whereas these results show that the muscle damage that occurs during pregnancy is partially reduced by Pilates. It is believed that the Pilates program conducted in this study improved muscle strength within the range that did not cause muscle damage.

References added are as follows. (line 523~536)

  1. Withee ED., et al., Effects of Methylsulfonylmethane (MSM) on exercise-induced oxidative stress, muscle damage, and pain following a half-marathon: a double-blind, randomized, placebo-controlled trial. J Int Soc Sports Nutr. 2017, Jul, 21;14:24.
  2. Liu Y., et al., AST to ALT ratio and arterial stiffness in non-fatty liver Japanese population:a secondary analysis based on a cross-sectional study. Lipids Health Dis. 2018 Dec 3;17(1):275.
  3. Pal S., et al., High-Intensity Exercise Induced Oxidative Stress and Skeletal Muscle Damage in Postpubertal Boys and Girls: A Comparative Study. J Strength Cond Res. 2018 Apr;32(4):1045-1052.
  4. Mannaerts D, et al., Oxidative stress in healthy pregnancy and preeclampsia is linked to chronic inflammation, iron status and vascular function. PLoS One. 2018. 11;13(9):e0202919.

--------------------------------------------------------------------------------

Point 5: Methods

No information has been provided in response to the questions about group allocation and randomisation. The response to this question was to say the study was a voluntary participation study, which is not an experimental design. In fact, all studies should be voluntary participation. Please provide the following information:

How were participants allocated to the experimental and control groups?

Were the assessors blinded to the group allocation?

Response 5: It has been modified as follows.  (line 104~117)

This study was a cluster randomized controlled experiment. A total of 16 subjects recruited were Pilates exercise group (PE, n=9; age:31.78±4.68 years, body weight (BW):, 57.97±7.91 kg, height: 160.56±3.78 cm, using the block randomization method). body fat mass (BFM): 18.40±4.92 g, BMI: 39.57±4.52, skeletal muscle mass (SMM): 21.10±2.66 g) and control (CON, n=7, age:32.00±3.46 years, BW:, 53.67 ±5.70 kg, height: 161.29±3.54 cm, BFM: 15.86±3.06 g, BMI: 37.81±3.37, SMM: 20.09±2.00 g) were randomly assigned.

In addition, the comparison of the reference values between groups was performed by obtaining Cohen's r values. The research protocol was approved by the Bioethics Committee of Korea National Sports University (1263-201803-BR-001-01).

For the explanation of the above block randomization method, refer to the paper below.

https://doi.org/10.4097/kja.19049

Chi-Yeon Lim, Junyong In., Randomization in clinical studies. 2019, Korean Journal of Anesthesiology, 72(3):221-232.

-------------------------------------------------------------------

Point 6: Results

A comparison of baseline values between groups is still required.

The comment in response to the request to transform Z into r values that can be interpreted as effect size measures was answered but no changes were made. These changes need to be made in a revised submission.

Response 6: We made a request to convert Z to an r value that can be interpreted as an effect size measure.

(line 186~192, line 222~223), (Table 2, Table 3, Table 4)
